# Immune Enhancement by the Tetra-Peptide Hydrogel as a Promising Adjuvant for an H7N9 Vaccine against Highly Pathogenic H7N9 Virus

**DOI:** 10.3390/vaccines10010130

**Published:** 2022-01-17

**Authors:** Xiaoxin Wu, Songjia Tang, Zhehua Wang, Xiaoyun Ma, Lingjian Zhang, Fen Zhang, Lanlan Xiao, Shuai Zhao, Qian Li, Ying Wang, Qingjing Wang, Keda Chen

**Affiliations:** 1State Key Laboratory for Diagnosis and Treatment of Infectious Diseases, Collaborative Innovation Center for Diagnosis and Treatment of Infectious Diseases, National Clinical Research Center for Infectious Diseases, The First Affiliated Hospital, Zhejiang University School of Medicine, Hangzhou 310003, China; xiaoxinwu@zju.edu.cn (X.W.); 11918236@zju.edu.cn (L.Z.); 11818053@zju.edu.cn (F.Z.); 13695732242@zju.edu.cn (L.X.); 3160105620@zju.edu.cn (S.Z.); leeqian@zju.edu.cn (Q.L.); 2Plastic and Aesthetic Surgery Department, Affiliated Hangzhou First People’s Hospital, Zhejiang University School of Medicine, Hangzhou 310000, China; tangsj@zju.edu.cn; 3Department of Infectious Disease and Medical Clinical Laboratory, Zhejiang Hospital, 1229 Gudun Road, Xihu, Hangzhou 310012, China; zhehuawang@yeah.net (Z.W.); yasmine919@163.com (X.M.); 4Shulan International Medical College, Zhejiang Shuren University, Hangzhou 310015, China

**Keywords:** immune enhancement, tetra-peptide hydrogel, adjuvant, H7N9 vaccine, highly pathogenic H7N9 virus

## Abstract

Background: Short peptide hydrogel was reported as a possible adjuvant for vaccines. In order to evaluate whether the Tetra-Peptide Hydrogel can be a promising adjuvant for an H7N9 vaccine against the highly pathogenic H7N9 virus, we conducted this study. Methods: Tetra-Peptide Hydrogels (D and L conformations) were prepared by a self-assembly system using a Naproxen acid modified tetra peptide of GFFY (Npx-GFFY). Mice received two immunizations with the D-Tetra-Peptide Hydrogel adjuvant vaccine, the L-Tetra-Peptide Hydrogel adjuvant vaccine, or the split vaccine. Fourteen days following the second dose, the mice were challenged with the highly pathogenic A/Guangdong/GZ8H002/2017(H7N9) virus. The mice were observed for signs of illness, weight loss, pathological alterations of the lung tissues and immune responses in the following 2 weeks. Results: The D/L-Tetra-Peptide Hydrogels resembled long bars with hinges on each other, with a diameter of ~10 nm. The H7N9 vaccine was observed to adhere to the hydrogel. All the unvaccinated mice were dead by 8 days post infection with H7N9. The mice immunized by the split H7N9 vaccine were protected against infection with H7N9. Mice immunized by D/L-Tetra-Peptide Hydrogel adjuvant vaccines experienced shorter symptomatic periods and their micro-neutralization titers were higher than in the split H7N9 vaccine at 2 weeks post infection. The hemagglutinating inhibition (HI) titer in the L-Tetra-Peptide Hydrogel adjuvant vaccine group was higher than that in the split H7N9 vaccine 1 week and 2 weeks post infection. The HI titer in the D-Tetra-Peptide Hydrogel adjuvant vaccine group was higher than that in the split H7N9 vaccine at 2 weeks post infection. Conclusion: The D/L Tetra-Peptide Hydrogels increased the protection of the H7N9 vaccine and could be promising adjuvants for H7N9 vaccines against highly pathogenic H7N9 virus.

## 1. Introduction

Throughout its history, mankind has fought against infectious diseases. Examples of infectious diseases experienced by humans include cholera; plague; smallpox; yellow fever; schistosomiasis; malaria; influenza; polio; whooping cough; diphtheria; measles; mumps; rubella; B encephalitis; epidemic meningitis; AIDS; syphilis; viral hepatitis; tuberculosis; Ebola virus disease; atypical pneumonia; Middle East respiratory syndrome; Zika virus disease; hand, foot and mouth disease; dengue fever; avian influenza; novel coronavirus disease and so on [1,2,3,4,5,6,7,8,9,10]. The control of infectious diseases mainly depends on controlling the source of infection, cutting off the transmission route, and protecting susceptible individuals. The most important measure to protect vulnerable groups is vaccination [10]. Vaccines play an important role in controlling infectious diseases. For example, smallpox was eliminated using a cowpox vaccine, and through vaccinations, humans were able to control polio, whooping cough, diphtheria, measles, mumps, rubella, B encephalitis, epidemic encephalitis, and tuberculosis [10,11,12]. Vaccines for hepatitis B; hepatitis A; hand, foot and mouth disease; influenza; and yellow fever have also played an important role in the control of the corresponding diseases [10,13,14,15,16]. Immunization has saved countless lives and is recognized as one of the most successful and cost-effective health interventions [13].

Among a number of different types of vaccines, inactivated adjuvant vaccines occupy a special place. Vaccine studies include the vaccine itself and the adjuvant. Research and development of vaccines has shown that some components can act as vaccine adjuvants, which can reduce the dosage and number of vaccinations, while enhancing the effects of humoral immunity and cellular immunity, and improving the effect of the vaccine on newborns, the elderly, and those with immunodeficiency [17,18]. Aluminum adjuvant is still the main adjuvant [18,19]. Adjuvants have the potential to increase local and systemic effects. Although many adjuvants work better than aluminum, they cannot be used for human vaccines because of their side effects [17,18,19]. Nanotechnology is a new technology, and nanoadjuvants have become a hot topic in vaccine research. Nanoadjuvants can absorb or wrap antigen particles to enhance their uptake by macrophages and dendritic cells (DCs). Short peptide hydrogels, formed by self-assembly of short peptides, have biological activity and good biocompatibility, and are easy to design and synthesize [20,21,22]. Nanofiber hydrogels enhance an animal’s immune response, activating both humoral immunity and cellular immunity, thus enabling DC cell activation and promoting the formation of germinal centers by increasing the intake of DC cell proteins [23]. The hydrogel (Naproxen acid (Npx)-GFFY) can activate cellular immune responses and humoral immune responses significantly. The hydrogel has the advantages of simple operation and mass preparation, and is an injectable physically interlinked hydrogel that can be used after uniform mixing with various antigens before use [24,25]. Therefore, the hydrogel can be used as an adjuvant for the vaccines and can be called hydrogel adjuvant. In previous studies, these hydrogel adjuvants had significant effects on all types of vaccines, such as those based on DNA, protein, tumor cell lysis solution, and short peptides [23,25,26]. The adjuvant itself can significantly activate the body’s immune system, improve the level of multiple cytokines (interleukin (IL)12, IL6, interferon (IFN)-γ, tumor necrosis factor alpha (TNFα)), and aid the development of vaccines against tumors, viruses, and other infectious agents [23]. 

The highly pathogenic H7N9 virus was reported to be able to infect humans in 2017, with an unacceptably high mortality of nearly 50% [27]. In addition, highly pathogenic H7N9 is already accumulating mutations, potentially increasing its infectivity toward humans, and thus causing a pandemic [28]. Fortunately, we successfully produced H7N9 virus vaccine strains using reverse genetics technology. The H7N9 split vaccine was deemed safe after the completion of active systemic anaphylaxis, repeated dose toxicity, and acute toxicity tests [29]. The H7N9 vaccine has also successfully completed immunogenicity and safety testing in an animal model [30]. The split H7N9 vaccine could protect BALB/c mice against highly pathogenic H7N9 virus infection; however, the protective effect decreased because of mutation of H7N9 [31]. Therefore, to enhance the activity of the H7N9 vaccine against highly pathogenic H7N9 virus, we developed Tetra-Peptide Hydrogel (D and L conformations) adjuvants. The present study was conducted to determine if the Tetra-Peptide Hydrogel (formed by Npx-GFFY) adjuvants could enhance the immunity of the H7N9 vaccine.

## 2. Materials and Methods 

### 2.1. Peptide Synthesis and Hydrogel Preparation 

Both Npx-G^D^F^D^F^D^Y and Npx-G^L^F^L^F^L^Y peptides were synthesized using routing Fmoc-solid peptide synthesis protocol with O-Benzotriazol-1-yl-tetramethyluronium hexafluorophosphate (HBTU) as the coupling activator and N,N-Diisopropylethylamine (DIPEA) as the activator base. D/L-isomeric Fmoc-protected amino acids were employed for D- and L-isomeric peptide synthesis, respectively. Meanwhile, the hydrogels were prepared using one heating–cooling protocol, as reported previously by Yang et al. [23,25]. In detail, 2 mg/mL peptide was dissolved in phosphate-buffered saline (PBS, pH = 7.4) and heated to approximately 95 °C. The system was left to cool down under ambient temperature; in this period, the hydrogel was formed.

### 2.2. Transmission Electron Microscopy (TEM)

The microscopic morphologies of the hydrogels (0.2%), the virus antigen (60 μg/mL) and the mixture of the hydrogels (0.2%) and the virus antigen (15 μg/mL) were observed using TEM. In general, 200-mesh copper grids were glow-discharged for 5 min and 10 μL samples were added onto the grids for 1 min. The excess liquid was removed using filter paper and the samples were stained using 3% uranium acetate for 1 min. Finally, the samples were dried using filter paper again and TEM images were collected under a TecnaiSpirit 120 kv Transmission Electron Microscope (Thermo Fisher Scientific, Waltham, MA, USA). 

### 2.3. Experimental Animal

Thirty specific-pathogen-free (SPF) female BALB/c mice (aged 6–8 weeks old) were provided by the Experimental Animal Center of Zhejiang Province, China. The animal experiments were carried out according to the Guide for the Care and Use of Laboratory Animals of Zhejiang Province. The Ethics Committee of the First Affiliated Hospital, Zhejiang University School of Medicine approved the animal study. All experiments with the high pathogenic H7N9 virus were performed in a China National Accreditation Service for Conformity Assessment-approved level 3 laboratory (Registration No. CNAS BL0022).

### 2.4. Cells and Viruses

The American Type Culture Collection (Manassas, VA, USA) provided the Madin-Darby canine kidney (MDCK) cell line. The cell lines were grown in Dulbecco’s modified Eagle medium supplemented with 10% fetal bovine serum in a 5% CO_2_ atmosphere at 37 °C. The A/Guangdong/GZ8H002/2017(H7N9) virus was isolated in 2017 from a patient in Guangdong Province, China. Allantoic cavities of 9-day-old SPF embryonated chicken eggs were used to propagate the virus for 72 h at 37 °C. A hemagglutinin (HA) assay was performed on the harvested allantoic fluid. In the assay, 50 μL of allantoic fluid solution was diluted to 1:2 using PBS and added to the wells of a 96-well blood coagulation plate. An equal volume of 1% chicken red blood cells was added to each well, and the plate was left at room temperature for 30–45 min. The HA titer was determined as the highest dilution at which blood coagulation occurred. Aliquots of virus-containing allantoic fluid were stored at −80 °C for further use.

### 2.5. Animal Immunization and Virus Inoculation

Groups (*n* = 6) of female BALB/c mice (19–21 g, *n* = 6) were immunized two times by intramuscular injection of 200 μL of the split vaccine alone (2.5 μg HA), the L-tetra-peptide hydrogel adjuvant vaccine (2.5 μg HA and 100 μL of L-Tetra-Peptide Hydrogel) or the D-tetra-peptide hydrogel adjuvant vaccine (2.5 μg HA and 100 μL of D-Tetra-Peptide Hydrogel). The negative and positive control group were immunized with PBS. The details of the groups are shown in Table 1.

On the day before virus incubation, mouse blood was sampled from the tail vein. Fourteen days after the boost dose, the mice were challenged with A/Guangdong/GZ8H002/2017(H7N9) virus at a TCID_50_ of 10^6^. Assaying the TCID_50_ was carried out according to a previously published protocol [30]. After infection, we monitored the mice for signs of weight loss, illness, and death. One week after virus challenge, a sample of the mice was euthanized humanely and their lungs and blood were collected (Figure 1). By eight days post infection, all the mice in the positive control group had died. The blood and lungs were collected from the dead mice. At 2 weeks post infection, all the surviving mice were euthanized humanely and their lungs and blood were collected. The lung tissues were divided into two parts, one of which was fixed in 10% buffered formalin, and the other subjected to TCID_50_ determination. Blood samples were subjected to centrifugation for 10 min at 500× *g* and the supernatant was retained as the serum, which was stored at −80 °C.

### 2.6. Histopathological Analysis of Lung Tissues

Hematoxylin eosin (HE) staining was performed on lung tissue sections. Mouse lung sections were then subjected to immunohistochemistry (IHC). We first dewaxed the paraffin-embedded lung tissue sections and heated them in citrate buffer. We quenched the endogenous peroxidase activity by incubating the sections in 0.3% H_2_O_2_ in methanol. Next, 3% bovine serum albumin (BSA) in PBS was used to block the sections for 1 h. The sections were then incubated with a 1:400 dilution of polyclonal rabbit antibodies against H7N9 at 4 °C overnight. Binding of the antibodies was detected employing the EnVision System (Agilent, Santa Clara, CA, USA). Hematoxylin counterstaining was carried out for all the slides.

### 2.7. Immunoglobulin G Enzyme-Linked Immunosorbent Assay (IgG-ELISA)

In PBS coating solution, 10 ng/well of H7N9 antigen was used to coat the wells of a 96-well polyvinyl chloride microtiter plate at 4 °C overnight. The wells were then incubated for 2 h with 3% BSA. After three PBS washes, 100 µl of a 2-fold serial dilution of serum (from 1:1000) was added to each well and incubated for 2 h. After five PBS washes, each well was added with 100 µL of a 1:10,000 dilution of peroxidase-labeled goat anti-mouse IgG and incubated for 2 h. The plates were washed and then 3,3′,5,5′-Tetramethylbenzidine (TMB) was added at 100 µL/well and incubated for 8 min, at which point the reaction was stopped. A plate reader was used to determine the absorbance in each well at wavelengths of 450 and 630 nm, and then the 450 nm/630 nm ratio was calculated after removing the average background OD value. Calculation showed that the ELISA value was 2.1-fold that of the average OD value of the negative control samples.

### 2.8. Hemagglutination Inhibition (HI) Titer Assay

The hemagglutination test was used to determine four units of the A/Guangdong/GZ8H002/2017(H7N9) virus. Before the HI titer tests, serum samples were heat-inactivated, diluted 1:10, and then a two-fold serial dilution series of the serum was added at a 1:1 ratio with four units of virus and incubated for 1 h at 37 °C. Thereafter, the samples were mixed with 1% chicken erythrocytes (50 µL) and incubated at room temperature for 1 h. The agglutination patterns of all the samples were read within 10 min.

### 2.9. Micro-Neutralization (MN) Assay

MDCK cells were seeded in 96-well plates at 2 × 10^4^ cells/well and incubated overnight at 37 °C until they reached 80–90% confluency. Serum samples were heat-inactivated, diluted with PBS by 1:10, serially diluted two-fold, and then added with A/Guangdong/GZ8H002/2017(H7N9) (100 TCID_50_ in 50 μL) and incubated at 37 °C for 2 h. Virus culture medium was then incubated with the cells for 48 h post-infection at 37 °C. ELISA was then used to detect the presence of the virus. 

The cells in all the wells were fixed with acetone for 20 minutes at room temperature. The acetone was removed and the wells were washed three times using PBS. The cells in the wells were then incubated for 2 h with 3% BSA. After three PBS washes, 100 µL of a 1:2000 dilution of the anti-H7N9 antibody was added to each well and incubated for 2 h. After five PBS washes, each well was added with 100 µL of a 1:10,000 dilution of peroxidase-labeled goat anti-mouse IgG and incubated for 2 h. The plates were washed and then TMB was added at 100 µL/well and incubated for 8 min, at which point the reaction was stopped. A plate reader was used to determine the absorbance in each well at wavelengths of 450 and 630 nm, and then the 450 nm/630 nm ratio was calculated after removing the average background OD value. Calculation showed that the ELISA value was 2.1-fold that of the average OD value of the negative control samples.

### 2.10. Statistical Analysis 

All statistical analyses were carried out using GraphPad Prism 5 software (GraphPad Inc., La Jolla, CA, USA). One-way analysis of variance (ANOVA) with Tukey’s multiple comparison test was carried out for the weight, IgG, HI and MN titer data. Data are shown as the mean ± SD for the indicated sample sizes. Statistical significance was indicated by a *p*-value < 0.05.

## 3. Results 

### 3.1. The Preparation of the Hydrogel Adjuvant Vaccine 

In the present study, we exploited the advantages of the Tetra-peptide hydrogel to develop a vaccine against highly pathogenic H7N9. We prepared hydrogels of Npx-GFFY in both D and L conformations using a routine heating-cooling protocol that was reported previously [23,25] (Figure 1A,B). As shown in Figure 1D,G, the H7N9 antigen (Split inactivated H7N9 virus) had a homogeneous but amorphous morphology, with a diameter reaching ~10 nm. Meanwhile, Npx-GFFY (D and L conformations) mainly consisted of uniform fibrils with a diameter of approximately 10 nm (Figure 1C,F). Reticular morphology was also observed, implying its higher-dimensional structure (Figure 1C,F). The hydrogel adjuvant vaccine was prepared by mixing the hydrogel and the H7N9 antigen together, and gently vortexing before injection. The H7N9 antigen was observed to adhere to the hydrogel (Figure 1E,H). The detailed compositions of the vaccines are listed in Table 1. 

### 3.2. The Protective Effects of the Hydrogel Adjuvant Vaccine In Vivo

The variation in body weights was used as the main index to evaluate mouse illness after H7N9 infection and the protective effects of the various vaccines (Table 1). As shown in Figure 1I, mice were immunized on days 0 and 14. At 14 days after the second dose, the mice in all the groups (Except the negative control group) were inoculated with A/Guangdong/GZ8H002/2017(H7N9) virus (hereafter abbreviated as H7N9) and weight variations for the next 14 days were recorded (Figure 2). H7N9 infection caused illness, weight loss, and death, and the hydrogel adjuvant vaccine had acceptable effects to reverse this process (Figure 2A–F). 

In the positive control group, the mice lost weight gradually over the first five days and sharply after the fifth day, reaching a weight loss of nearly 20% before death (Figure 2A, Appendix A). A third (33%) of the mice had died by day 7 post infection (Figure 2B). The remaining live mice presented signs of severe respiratory disease, including lack of appetite and respiratory distress. In the positive control group, by 8 days post infection, all the mice had died (Figure 2B). By contrast, the body weights of the uninfected mice (Negative control group) increased by 14.4 ± 3.2% over 7 days (Figure 2A,F). Interestingly, both vaccines showed some ability to shorten the weight loss period and enable the recovery of body weight after 6 days post-infection (Figure 2A). The hydrogel adjuvant vaccine possessed the greatest protective effects against H7N9 infection among the groups. The weights of the mice in the L-hydrogel vaccine group were not significantly lower than those in the negative group at one week after virus inoculation (Figure 2A–F). The weights of the mice in the D-hydrogel vaccine group were significantly lower than those in the negative group at 4 days after virus inoculation (Figure 2D); however, the weights of the mice in the D-hydrogel vaccine group increased and were not significantly lower than those in the negative group at 6 and 7 days after virus inoculation (Figure 2E,F).

In the following days, the weights of mice in the vaccine groups were lower than those of the negative group; however, the weight of mice in the H7N9 vaccine and hydrogel vaccine groups recovered to that before incubation with the virus (Figure 2G–I). Meanwhile, the weights of the mice in the L-hydrogel vaccine group were similar to those in the negative group at 2 weeks after virus infection (Figure 2I).

### 3.3. The Protective Effects of Hydrogel Adjuvant Vaccine against Lung Damage 

The respiratory distress and lung damage of the mice infected with H7N9 is one of the main clinical signs. As shown in Figure 1I, at 1 week after infection with the H7N9 virus, mice were sacrificed and their lung tissues were evaluated using both hematoxylin-eosin (HE) and immunohistochemical (IHC) staining to reveal the effects of the hydrogel adjuvant vaccine against lung infection and damage from H7N9 (Figure 3). Compared with uninfected mice (negative control), the lungs of the H7N9-only group (positive control) showed exudative pathological alterations and multifocal interstitial inflammatory hyperemia. A high infection level was observed in the mice treated using H7N9 only (positive control), especially in the region of bronchiolar epithelium (IHC panel). We clearly observed that all vaccines could prevent the H7N9 infection and lung tissue damage, appearing as lower levels of hollow-like structures (HE panel) and no obvious viral antigens were detected (IHC panel). No live virus was detected in the lung tissue leaching solution of mice in the vaccine groups (Appendix A). Taken together, these results suggested that the hydrogel adjuvant vaccine could effectively protect the lung from the virus infection and potential damage. 

### 3.4. The Hydrogel Adjuvant Vaccine Elicited Good Protective Antibody Titers against H7N9 Virus

To evaluate the immune response of the vaccines, we conducted series of measurements of antibody titers via IgG, HI, and MN assays. Sera were collected on day 28 after first immunization and on day 7 and day 14 after H7N9 infection, respectively. The results showed that both vaccines could elicit high IgG titers against H7N9 after the second boost (Figure 4A–C). The IgG titers against H7N9 continued to increase in the hydrogel vaccine group (Figure 4A–C). At two weeks after the second dose, the HI titers in the H7N9 vaccine and hydrogel vaccine groups were similar (Figure 4D). At 1 week and 2 weeks after challenge, the HI titer of the mice in the L-hydrogel vaccine group was higher than that in the H7N9 vaccine group (Figure 4E,F). Meanwhile, at 1 week after challenge, the HI titer of the mice in the L-hydrogel vaccine group was higher than that in the D-hydrogel vaccine group (Figure 4E). However, at 2 weeks after challenge, the HI titers of the mice in the L-hydrogel vaccine and the D-hydrogel vaccine groups were similar (Figure 4F). Furthermore, the HI titer of the mice in the D-hydrogel vaccine group was higher than that in the H7N9 vaccine group at 2 weeks after infection.

At two weeks after the second dose and 1 week after infection, the MN titers in the H7N9 vaccine and the hydrogel vaccine group were similar (Figure 4G–H), while the MN titer of the mice in the hydrogel vaccine groups was higher than that in the H7N9 vaccine at 2 weeks after virus challenge (Figure 4I). Use of the hydrogel adjuvant vaccine resulted in the highest HI and MN titers, suggesting that the hydrogel adjuvant vaccine induced strong immune memory and confirmed the importance of adjuvant selection for antiviral vaccine development.

## 4. Discussion

Vaccines are currently the most effective and economical way to combat human infectious diseases [13]. There are various types of vaccines, including live attenuated vaccines, inactivated vaccines, carrier vaccines, mRNA vaccines, recombinant protein vaccines, and DNA vaccines [32,33]. The technology to produce inactivated vaccines is mature and their production is stable. Inactivated vaccines are currently the most mainstream vaccines on the market and are available for both influenza virus and coronavirus [33,34]. Inactivated vaccines are relatively safe; however, the levels of antibodies induced by them are relatively lower than those produced by live attenuated vaccines and mRNA vaccines, especially the influenza inactivated vaccine and COVID-19 inactivated vaccine [35]. Adjuvants play an important role in vaccine immunity [17], and research into adjuvants is a hot topic, because adjuvants can reduce the use of the immunogenic components of the vaccine and also significantly increase the yield and immune effects [18]. In this context, nanoadjuvants represent a new type of adjuvant.

Among various nanomaterials, the self-assembly of peptides into supramolecular hydrogels via noncovalent interactions has become a research hotspot because of their facile design, known functional motifs, low immunogenicity, biocompatibility, and biodegradability [25,26,36]. Many hydrogels based on short peptides have been developed, which demonstrate marked potential in sensing applications, drug delivery, three-dimensional cell culture, immunomodulation, and cancer cell inhibition [36]. Moreover, short peptide-based hydrogels are powerful immune adjuvants and, thus, might be formulated into potent systems for antigen delivery [24,26,36,37]. The hydrogel adjuvant is a new type of adjuvant, which, via its antigen components, can increase the intake of protein, activate DC cells, promote the formation of germinal centers, and slowly release the antigen components, thereby achieving a continuous stimulation effect and producing a higher antibody titer [23,38]. As an important reserve vaccine, the H7N9 vaccine is used to deal with possible future H7N9 epidemics [28]. At present, the H7N9 vaccine itself has been fully studied; however, it remains important to develop an adjuvant with fewer side effects and higher efficacy [30,31]. There are few relevant studies on whether a hydrogel adjuvant can increase the immune effects of the H7N9 vaccine and there are no data on whether a hydrogel could enhance the effects of the H7N9 vaccine against infection with highly pathogenic H7N9. In our study, we prepared the D/L-Tetra-Peptide hydrogel adjuvants that have been shown to be used for other vaccines [24,25]. In our experiment, the H7N9 vaccine antigen component was small in diameter. We found that hydrogel adjuvants can carry H7N9 antigen, increase the diameter of the H7N9 antigen, and can more effectively stimulate the immune system. H7N9 antigen can be released slowly and exert lasting stimulation effects.

The results of the present study showed that without vaccine protection, the highly pathogenic H7N9 virus was fatal and almost all the mice died within eight days, which was similar to our previous studies [27,31]. The H7N9 split vaccine protected the mice; however, they experienced a transient period of weight loss and began to regain weight after 6 days. The hydrogel adjuvant vaccine groups had small lesions after highly pathogenic H7N9 challenge and experienced a short period of weight loss. Both the H7N9 vaccine group and hydrogel adjuvanted vaccinated groups had no breakthrough infections. In the positive control group, the mice will have a relatively stable time in the first few days after infection with the virus, and later suddenly enter a period of acceleration, rapidly develop into a poor state as weight loss, and finally death. This phenomenon is also common in clinical work and in our previous study [27]. The most severe clinical signs were observed at 6–8 days after the infection. In this period, the weight of the mice represents the status of the mice. On the sixth and seventh day after infection, the weights of the mice in the H7N9 vaccine group were significantly lower than the negative control group. However, there were no significant differences between the hydrogel adjuvanted vaccinated groups and the negative control group. These data also support that the hydrogel adjuvanted vaccines are an improvement on the H7N9 vaccine to some extent. On the seventh day after infection, the average weight of the H7N9 vaccine group was also lower than that of the hydrogel adjuvanted vaccinated group; however, the difference was not statistically significant. At the same time, we found that although the vaccine had a protective effect, the vaccinated mice still could not achieve the weight of the normal mice, indicating that viral challenge still affected the growth of the vaccinated mice.

In our study, both vaccines could elicit high IgG titers against H7N9 after the second boost, but the IgG titers of the H7N9 vaccine group were higher than the hydrogel vaccine groups. However, the IgG titers against H7N9 continued to increase in the hydrogel vaccine groups. The IgG titers of the D-hydrogel vaccine group were higher than the H7N9 vaccine group 2 weeks after virus infection. The hydrogels can absorb or wrap antigen particles, and release antigen slowly and effectively, thus persistently stimulating the immune response [20,21,22,23,24,25,26]. In studies of influenza vaccines, IgG antibodies represent total antibodies, and the antibodies we tested represent various protein-type antibodies. However, total IgG antibodies do not represent true protective antibodies. In influenza research, blood coagulation inhibition antibodies and microneutralizing antibodies can truly respond to the effect of the vaccine [39]. Our study also confirmed that hydrogel adjuvant vaccines produce higher levels of HI and microneutralizing antibodies that are more protective to mice. In previous studies, hydrogels increased antibody efficiency, which was confirmed in these experiments [23,25,40,41]. Our research was conducted on inactivated vaccines, and we demonstrated that our hydrogel vaccine improved the effectiveness of H7N9 inactivated vaccines. Currently, the new coronavirus inactivated vaccine is one of the main vaccines used in the fight against COVID-19, and our hydrogel might also increase the effectiveness of new coronavirus inactivated vaccines, and thus, related issues are worthy of further research.

There are also limitations to our study. First, because we used a limited number of animals, the results of our study may fail to fully demonstrate the effect of the Tetra-Peptide Hydrogel. Second, we only detected the humoral immune effect of each group, and cellular immunity was not explored in this study. Third, the mechanism underlying the adjuvant’s role was not studied specifically. 

## 5. Conclusions

In summary, the results showed that that the coassembly system-formed hydrogel-induced strong antibody production when combined with the H7N9 vaccine. The Tetra-Peptide Hydrogel (D and L conformations) could be a promising adjuvant for inactivated vaccines. 

## Figures and Tables

**Figure 1 vaccines-10-00130-f001:**
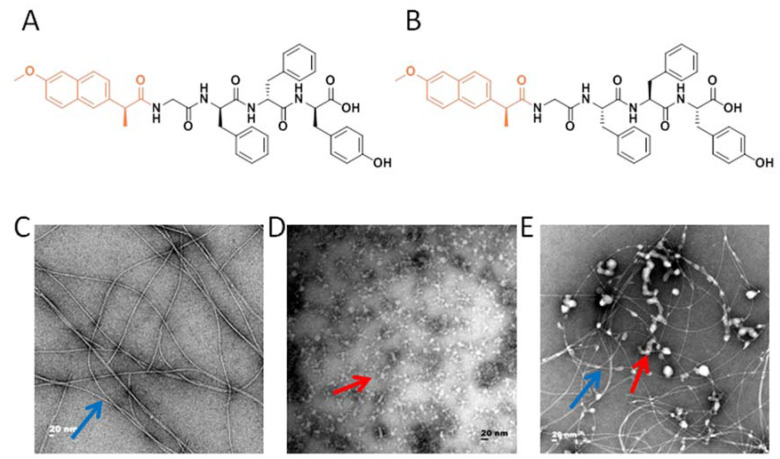
The microstructure of the adjuvant vaccine and the schematic illustration of immunization procedure. (**A**) The molecular structure of Npx-G^D^F^D^F^D^Y (D conformation). (**B**) The molecular structure of Npx-G^L^F^L^F^L^Y (L conformation). (**C**) Transmission electron microscopy (TEM) photomicrographs of the D-Tetra-Peptide Hydrogel. (**D**) TEM photomicrographs of the H7N9 antigen. (**E**) TEM photomicrographs of the mixture of D-Tetra-Peptide Hydrogel and vaccine. (**F**) TEM photomicrographs of the L-Tetra-Peptide Hydrogel. (**G**) TEM photomicrographs of the H7N9 antigen. (**H**) TEM photomicrographs of the mixture of L-Tetra-Peptide Hydrogel and vaccine. (**I**) Schematic illustration of immunization procedure. Six mice in each group were immunized for further evaluation. The representative morphology of the vaccine antigen is indicated with a red arrow and the representative morphology of the hydrogel is indicated with a blue arrow.

**Figure 2 vaccines-10-00130-f002:**
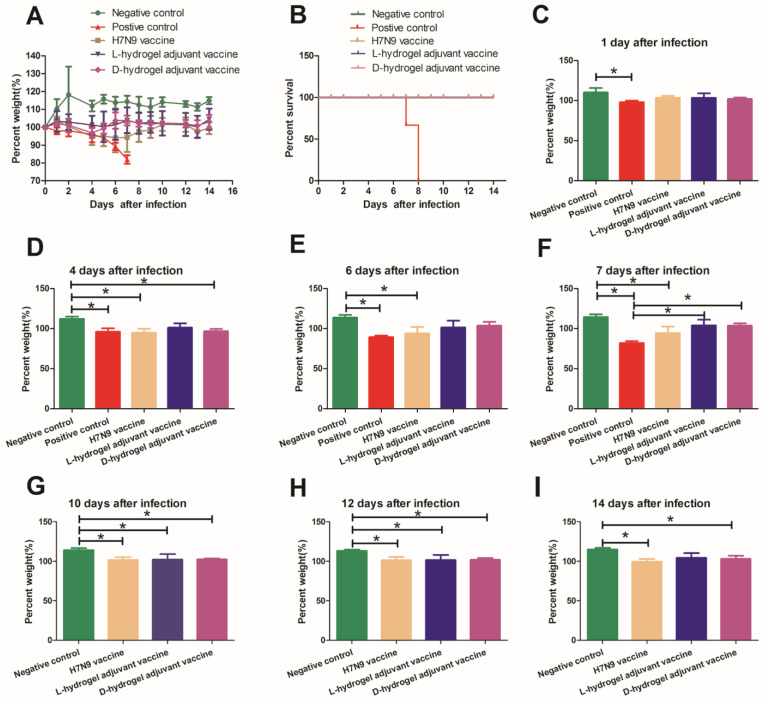
Weights and survival of mice infected with the H7N9 virus and treated with various vaccines. (**A**) Weight variations of mice infected with H7N9 virus (50 µL 10^6^ TCID50). Groups of mice were immunized with the indicated vaccines on day 0 and 14 before intranasal virus challenge. The percent weight was calculated as the percentage of weight on indicated days versus the weight on initial day (the day when the mice were inoculated with H7N9 virus). (**B**) The survival curve of the five groups. (**C**–**I**) The percent weight calculated at 1, 4, 6, 7, 10, 12, and 14 days after infection, respectively. Data are shown as the mean ± SD for the indicated sample sizes. The mean values with error bars representing one standard deviation. An asterisk (*) represents a *p* value < 0.05.

**Figure 3 vaccines-10-00130-f003:**
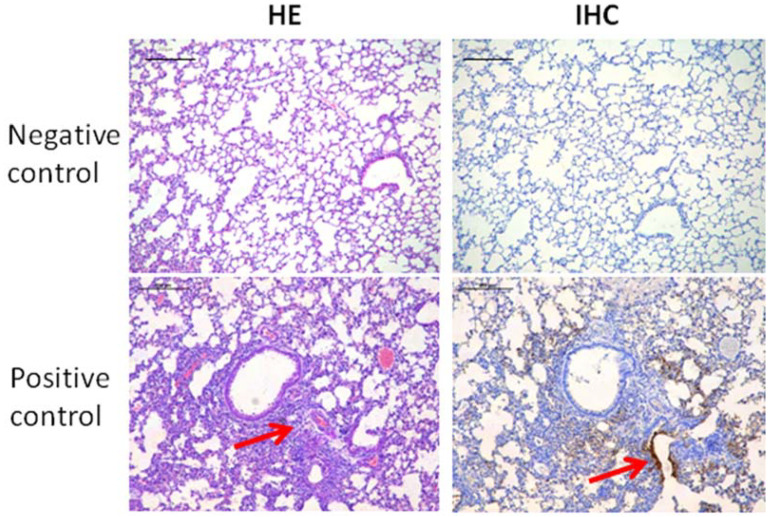
Representative HE (**left** panel) and IHC (**right** panel) staining images of lung tissues after various treatments. A hollow-like structure (red arrow) represents the tissue damage (**left** panel). Tissue highlighted by polyclonal rabbit anti-H7N9 antibody reflecting the levels of H7N9 infection (**right** panel). The scale bars in HE and IHC staining denote 200 μm.

**Figure 4 vaccines-10-00130-f004:**
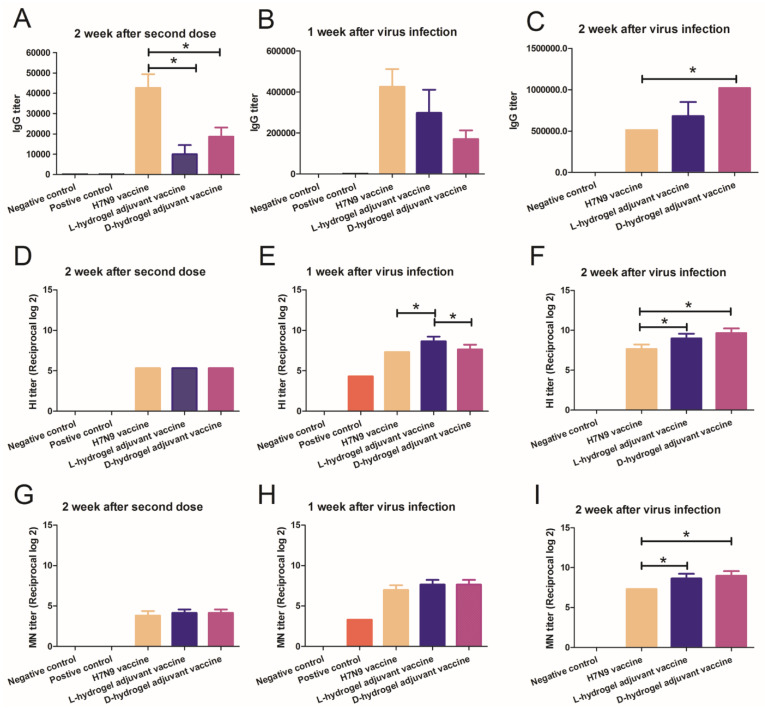
IgG, HI, and MN titers in immunized mice. IgG titer measurements of in the sera of mice immunized with different agents at 2 weeks after the second dose (**A**), at 1 week post virus inoculation (**B**), and at a further 2 weeks post virus inoculation (**C**). HI titer measurements in the sera of mice immunized with different agents at 2 weeks after the second dose (**D**), at 1 week post virus inoculation (**E**), and at a further 2 weeks post virus inoculation (**F**). MN titer measurements of sera of mice immunized with different agents at 2 weeks after the second dose (**G**), at 1 week post virus inoculation (**H**), and at a further 2 weeks post virus inoculation (**I**). An asterisk (*) represents a *p* value < 0.05.

**Table 1 vaccines-10-00130-t001:** Experiment grouping design.

Group Name	Vaccine Composition	HA	Adjuvant	Virus Inoculation
Negative control	PBS	−	−	−
Positive control	PBS	−	−	A/Guangdong/GZ8H002/2017(H7N9) virus
H7N9 vaccine	H7N9 vaccine	2.5 μg/mouse	−	A/Guangdong/GZ8H002/2017(H7N9) virus
L-hydrogel adjuvant vaccine	L-Tetra-Peptide Hydrogel and H7N9 vaccine	2.5 μg/mouse	100 μL/mouse	A/Guangdong/GZ8H002/2017(H7N9) virus
D-hydrogel adjuvant vaccine	D-Tetra-Peptide Hydrogel and H7N9 vaccine	2.5 μg/mouse	100 μL/mouse	A/Guangdong/GZ8H002/2017(H7N9) virus

“−”denotes no corresponding content was included. PBS, phosphate-buffered saline; HA, hemagglutinin.

## Data Availability

The data presented in this study are available on request from the corresponding author.

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
