# Peer review of "Immune Enhancement by the Tetra-Peptide Hydrogel as a Promising Adjuvant for an H7N9 Vaccine against Highly Pathogenic H7N9 Virus"

_vaccines, 2022, doi:10.3390/vaccines10010130_

Round 1

Reviewer 1 Report

This is a very interesting and timely work that sounds like a proof of concept. The D/L Tetra-Peptide Hydrogels could be promising adjuvants not only for vaccines against highly pathogenic influenza viruses but for any vaccine for human use. I think that for non-respiratory diseases such adjuvants could be especially important.

Point 1: Line 18-19. The sentence “To explore the immune…” is not the BACKGROUND but the PRIMARY GOAL. Please, present the BACKGROUND correctly. 

Point 2: Line 24-27. Too many details for the ABSTRACT. Вполне можно ограничиться заменой “For the next 2 weeks, we observed the mice for signs of illness and weight loss. At 1 week post-infection, a number of mice in each group were sacrificed and their lungs were excised. The pathological alterations to the lung tissues and the immune responses were assessed.” With “The mice weку observed for signs of illness, weight loss, the pathological alterations to the lung tissues and the immune responses.”

Point 3: Line 60. There is no logical transition from the above paragraph to this paragraph. Before the sentence “Vaccine studies include the vaccine itself and the adjuvant”, it should be briefly said that, for example, “Among a number of different types of vaccines, inactivated adjuvant vaccines occupy a special place”.

Point 4: Line 87-88. Please, specify here that the vaccine you have developed was split one.  

Point 5: Line 132. Please, explain why did you propagate the H7N9 virus in eggs for 72 hours? 48 hours are more than enough for the avian influenza A virus. 

Point 6: Line 133-138. I suppose that there is no necessity to describe in such small details routine HA assay. It will be enough to refer, for instance, to WHO Manual for the laboratory diagnosis and virological surveillance of influenza.

Point 7: Line 139. Please, do not capitalize “Virus” 

Point 8: Table 1, Figures. This is confusing and not clear, did you use the only split vaccine in all experimental groups, or split vaccine was used as control only, and preparations titled “adjuvant vaccine” contained another type of vaccine. Please, uniform group names. Maybe, it is better to delete the word “split” from the group 3 name.

Point 9: Table 1, Figures, etc. “Normal control” is not the best name for this group. It should be renamed as “Negative control.”

Point 10: Line 192-198. I suppose that there is no necessity to describe in detail routine HI assay (see point 6). It will be enough to refer, for instance, to WHO Manual for the laboratory diagnosis and virological surveillance of influenza.

Point 11: Line 225-228. This sentence belongs to the Introduction. Please, move it there. In general, a brief introduction from 1-2 sentences should be added to the Introduction section describing what does it mean - hydrogel adjuvants.

Point 12: Did you perform a safety study on your Tetra-Peptide Hydrogel adjuvants? Please, add this information.

Author Response

This is a very interesting and timely work that sounds like a proof of concept. The D/L Tetra-Peptide Hydrogels could be promising adjuvants not only for vaccines against highly pathogenic influenza viruses but for any vaccine for human use. I think that for non-respiratory diseases such adjuvants could be especially important.

Point 1: Line 18-19. The sentence “To explore the immune…” is not the BACKGROUND but the PRIMARY GOAL. Please, present the BACKGROUND correctly. 

Response: We would like to thank you for your thoughtful and detailed review and consideration of our manuscript. The comments and suggestions made by you have helped us improve the manuscript significantly. We have revised it according to you recommendation. The change has been highlighted in the revised manuscript. We have provided the BACKGROUND correctly. 

Point 2: Line 24-27. Too many details for the ABSTRACT. Вполне можно ограничиться заменой “For the next 2 weeks, we observed the mice for signs of illness and weight loss. At 1 week post-infection, a number of mice in each group were sacrificed and their lungs were excised. The pathological alterations to the lung tissues and the immune responses were assessed.” With “The mice weку observed for signs of illness, weight loss, the pathological alterations to the lung tissues and the immune responses.”

Response: We would like to thank you for your thoughtful and sincere review. The comments and suggestions made by you have helped us improve the manuscript. We have revised our abstract section. The change has been highlighted in the revised manuscript.

Point 3: Line 60. There is no logical transition from the above paragraph to this paragraph. Before the sentence “Vaccine studies include the vaccine itself and the adjuvant”, it should be briefly said that, for example, “Among a number of different types of vaccines, inactivated adjuvant vaccines occupy a special place”.

Response: Thank you so much. The comments and suggestions made by you have helped us improve the manuscript significantly. We have revised it according to you recommendation.

Point 4: Line 87-88. Please, specify here that the vaccine you have developed was split one.  

Response: Thank you very much. We have revised it according to you suggestion.

Point 5: Line 132. Please, explain why did you propagate the H7N9 virus in eggs for 72 hours? 48 hours are more than enough for the avian influenza A virus. 

Response: We would like to thank you for your thoughtful and sincere review. You're right, however we still need to provide the real culture conditions in this study. In our laboratory, we routinely cultured the avian influenza A virus for 72 hours.

Point 6: Line 133-138. I suppose that there is no necessity to describe in such small details routine HA assay. It will be enough to refer, for instance, to WHO Manual for the laboratory diagnosis and virological surveillance of influenza.

Response: We would like to thank you for your thoughtful and sincere review. Of course, referring is enough to some extent, however we also hope that readers could quickly understand each step of our experimental operation, which is more convenient for the readers.

Point 7: Line 139. Please, do not capitalize “Virus” 

Response: Thank you very much. We have revised it

Point 8: Table 1, Figures. This is confusing and not clear, did you use the only split vaccine in all experimental groups, or split vaccine was used as control only, and preparations titled “adjuvant vaccine” contained another type of vaccine. Please, uniform group names. Maybe, it is better to delete the word “split” from the group 3 name.

Response: Thank you very much. We would like to thank you for your thoughtful and sincere review. We have revised it according to you recommendation.

Point 9: Table 1, Figures, etc. “Normal control” is not the best name for this group. It should be renamed as “Negative control.”

Response: We would like to thank you for your thoughtful and sincere review. We have revised it according to you suggestion.

Point 10: Line 192-198. I suppose that there is no necessity to describe in detail routine HI assay (see point 6). It will be enough to refer, for instance, to WHO Manual for the laboratory diagnosis and virological surveillance of influenza.

Response: We would like to thank you for your thoughtful and sincere review. You're right, but we hope that readers can quickly understand each step of our experimental operation, which is more convenient for all the readers.

Point 11: Line 225-228. This sentence belongs to the Introduction. Please, move it there. In general, a brief introduction from 1-2 sentences should be added to the Introduction section describing what does it mean - hydrogel adjuvants.

Response: Thank you very much. We have revised it according to you recommendation. The change has been highlighted in the revised manuscript.

Point 12: Did you perform a safety study on your Tetra-Peptide Hydrogel adjuvants? Please, add this information.

Response: We did not do the safety experiments in this experiment. We will complete the safety research before applying it to people, which will be our subsequent research direction.

Reviewer 2 Report

The authors present an interesting study where they have explored the use of Tetra-Peptide Hydrogels as adjuvants to enhance the protective effects of an inactivated H7N9 vaccine in a mouse model. The manuscript is generally well written, and the results are generally well presented.

The main problem with this manuscript is the discussion section. I believe this section requires a complete rewrite. The first couple of paragraphs (lines 337-367) are very general in nature and could arguably be considered introductory text. The third and fourth paragraphs mention the project to some extent:

“Our research was conducted on inactivated vaccines, and we demonstrated that our hydrogel vaccine improved the effectiveness of H7N9 inactivated vaccines.”

However, I am not sure the presented data supports this conclusion. To support this conclusion the authors need to sequentially the presented data (evidence) to show this is valid. It is certainly true that the hydrogel adjuvanted vaccines elicit protective responses. What I would ask the authors is what is the evidence that the hydrogel adjuvanted vaccines are an improvement on their existing split H7N9 vaccine?

For example, how does the data shown in Fig 2c to 2I support this? No significant differences were detected between the split vaccine group and either of the hydrogel adjuvanted vaccinated groups. Then moving on to Fig 3 – does this support an improvement in the vaccine when hydrogel adjuvanted?

Then we moved Fig 4 – immune responses. How do these results correlate with the previously presented data? If they do, what does this mean? If one (or more) groups are responding better, did this equate improve protection from a clinical signs point of view?

other interesting question not addressed, but could be considered are:

  • Which is better, if any, in respect to the L-hydrogel and the R-hydrogel adjuvanted vaccines?
  • Why were different isomers used in this context?
  • How does this fit previous literature on hydrogels?

Line 60 suggest deletion of “Vaccine studies include the vaccine itself and the adjuvant.” as I am not sure of its value or actual intended message, vaccine studies encompass far more than this cursory statement and not all vaccines contain adjuvants.

Line 66-67 Please provide an appropriate citation(s) for this statement regarding the use of alum based adjuvants.

Line 75 suggest revision “can significantly activate cellular immune and humoral immune responses.”

“immunity” and “immune responses” are not the same thing.

Line 81 Would the balance of the increase in these cytokines also be important?

Line 84 suggest revision “was reported to infected humans”

Line 89 Please provide a reference for this statement of prior data publication.

Line 141 – can the authors confirm that no adjuvant was used in the split vaccine formulation?

Line 241 the figure legend should have a general title to indicate what the figure illustrates, before describing the individual panels.

Line 261 suggest revision “had signs of severe respiratory disease,”

Line 254 please review this text, according to Table 1, not “all groups were inoculated”

Line 287 suggest revision “is one of the main clinical signs.”

Line 280 Figure 2 –

Figure 2A -

Panel B I gather that the data for most of the groups are superimposed on each other as it is not very informative in terms of data visibility. Perhaps consider providing a graph for each group as a supplemental figure.

The figure legend should include a description of what data is shown, for example the mean values with error bars representing one standard deviation or standard error of the mean.

Line 282 Please review this text, I do not think the mice were immunised every day for 14 days.

Line 301 Figure 3 – there appears to be antigen staining in the IHC images for the Split Vaccine and the L-hydrogel adjuvant vaccine groups but these are no “indicated” arrows. Please review.

Line 328 Figure 4 - I would suggest replacing “attack” with “challenge” or “infection”. I would also suggest having the Y-axis scale the same for Panels D, E, & F and also for Panels G, H, and I – this would enable easier comparison of these results.

Author Response

The authors present an interesting study where they have explored the use of Tetra-Peptide Hydrogels as adjuvants to enhance the protective effects of an inactivated H7N9 vaccine in a mouse model. The manuscript is generally well written, and the results are generally well presented. The main problem with this manuscript is the discussion section. I believe this section requires a complete rewrite. The first couple of paragraphs (lines 337-367) are very general in nature and could arguably be considered introductory text. The third and fourth paragraphs mention the project to some extent:

 Response: We would like to thank you for your thoughtful and detailed review and consideration of our manuscript. The comments and suggestions made by you have helped us improve the manuscript significantly. We have rewritten the discussion section. The change has been highlighted in the revised manuscript.

“Our research was conducted on inactivated vaccines, and we demonstrated that our hydrogel vaccine improved the effectiveness of H7N9 inactivated vaccines.”

However, I am not sure the presented data supports this conclusion. To support this conclusion the authors need to sequentially the presented data (evidence) to show this is valid. It is certainly true that the hydrogel adjuvanted vaccines elicit protective responses. What I would ask the authors is what is the evidence that the hydrogel adjuvanted vaccines are an improvement on their existing split H7N9 vaccine?

 Response: We would like to thank you for your thoughtful and detailed review and consideration of our manuscript. The comments and suggestions made by you have helped us improve the manuscript significantly. Mice immunized by D/L-Tetra-Peptide Hydrogel adjuvant vaccines experienced shorter symptomatic periods and their micro‑neutralization titers were higher than in the split H7N9 vaccine at 2 weeks post infection. The hemagglutinating inhibition (HI) titer in L-Tetra-Peptide Hydrogel adjuvant vaccine group was higher than that in the split H7N9 vaccine a 1 week and 2 weeks post infection. The HI titer in D‑Tetra-Peptide Hydrogel adjuvant vaccine group was higher than that in the split H7N9 vaccine at 2 weeks post infection. In influenza research, blood coagulation inhibition antibodies and microneutralizing antibodies can truly respond to the effect of the vaccine.

For example, how does the data shown in Fig 2c to 2I support this? No significant differences were detected between the split vaccine group and either of the hydrogel adjuvanted vaccinated groups. Then moving on to Fig 3 – does this support an improvement in the vaccine when hydrogel adjuvanted?

Then we moved Fig 4 – immune responses. How do these results correlate with the previously presented data? If they do, what does this mean? If one (or more) groups are responding better, did this equate improve protection from a clinical signs point of view?

Response: We would like to thank you for your thoughtful and detailed review and consideration of our manuscript. In the positive control group, we could clearly observe that 6-8 days after infection were the most important time. In this period, the weight of mice represented the status of the mice. On 6 days and 7 days after infection, the weights of mice were of no significant difference between the hydrogel adjuvanted vaccinated groups and the normal group. These data also indicated that the hydrogel adjuvanted vaccines are an improvement on H7N9 vaccine. Indeed, the Figure 3 is difficult to support the improvement in the adjuvant vaccines, but at least it indicated that the adjuvant vaccine is effective. The mice immunized by split H7N9 vaccine were protected from infection with H7N9. Mice immunized by D/L-Tetra-Peptide Hydrogel adjuvant vaccines experienced shorter symptomatic periods and their micro‑neutralization titers were higher than in the split H7N9 vaccine at 2 weeks post infection. Better immune responses and shorter symptomatic periods did equate the improved protection from a clinical signs point of view.

other interesting question not addressed, but could be considered are:

  • Which is better, if any, in respect to the L-hydrogel and the R-hydrogel adjuvanted vaccines?
  • Why were different isomers used in this context?
  • How does this fit previous literature on hydrogels?

 Response: We would like to thank you for your thoughtful and detailed review and consideration of our manuscript. The comments and suggestions made by you have helped us improve the manuscript significantly. In our study, the D/L Tetra-Peptide Hydrogels increased the protection of the H7N9 vaccine and could be promising adjuvants for H7N9 vaccines against highly pathogenic H7N9 virus. We think both of them work. The D and L conformation was extensively existed. We used different isomers in this study in order to clarify if there is a big difference between them. In the previous studies, the D conformation was reported to be better, but in our study, we think both of them work. There is no significance between them.

Line 60 suggest deletion of “Vaccine studies include the vaccine itself and the adjuvant.” as I am not sure of its value or actual intended message, vaccine studies encompass far more than this cursory statement and not all vaccines contain adjuvants.

Response: Thank you very much. We would like to thank you for your thoughtful and sincere review. We have added this sentence " Among a number of different types of vaccines, inactivated adjuvant vaccines occupy a special place " before the sentence "Vaccine studies include the vaccine itself and the adjuvant”. The change has been highlighted in the revised manuscript.

Line 66-67 Please provide an appropriate citation(s) for this statement regarding the use of alum based adjuvants.

Response: Thank you very much. We have revised it according to you recommendation.

Line 75 suggest revision “can significantly activate cellular immune and humoral immune responses.”“immunity” and “immune responses” are not the same thing.

Response: Thank you very much. We would like to thank you for your thoughtful and sincere review. We have revised it according to you recommendation.

Line 81 Would the balance of the increase in these cytokines also be important?

Response: Thank you for your comment. Adjuvants can improve the level of these cytokines. Of course, the balance of the increase in these cytokines is also very important.

Line 84 suggest revision “was reported to infected humans”

Response: Thank you very much. We have revised it according to you recommendation.

Line 89 Please provide a reference for this statement of prior data publication.

Response: Thank you very much. We would like to thank you for your thoughtful and sincere review. We have provided a reference for this statement of prior data publication.

Line 141 – can the authors confirm that no adjuvant was used in the split vaccine formulation?

Response: We confirmed that no adjuvant was used in the split vaccine formulation in this study.

Line 241 the figure legend should have a general title to indicate what the figure illustrates, before describing the individual panels.

Response: Thank you very much. We have revised it according to you recommendation. The figure legend would be " The microstructure of the adjuvant vaccine and the schematic illustration of immunization procedure ".

Line 261 suggest revision “had signs of severe respiratory disease,”

Response: Thank you very much. We have revised it according to you recommendation.

Line 254 please review this text, according to Table 1, not “all groups were inoculated”

Response: Thank you very much. We would like to thank you for your thoughtful and sincere review. We have revised it in the revised manuscript. The change has been highlighted in the revised manuscript.

Line 287 suggest revision “is one of the main clinical signs.”

Response: Thank you very much. We have revised it in the revised manuscript.

Line 280 Figure 2 – Figure 2A -Panel B I gather that the data for most of the groups are superimposed on each other as it is not very informative in terms of data visibility. Perhaps consider providing a graph for each group as a supplemental figure. The figure legend should include a description of what data is shown, for example the mean values with error bars representing one standard deviation or standard error of the mean.

Response: Thank you very much. The percent weight calculated at 1, 4, 6, 7, 10, 12, and 14 days after infection were presented in figure 2C-2I respectively. We have provided the graph for each group as a supplemental figure 1. Meanwhile, in the figure legend, we added a description of what data is shown. " Data are shown as the mean ± SD for the indicated sample sizes. The mean values with error bars representing one standard deviation."

Line 282 Please review this text, I do not think the mice were immunised every day for 14 days.

Response: Thank you very much. It is a mistake. We have revised it in the revised manuscript.

Line 301 Figure 3 – there appears to be antigen staining in the IHC images for the Split Vaccine and the L-hydrogel adjuvant vaccine groups but these are no “indicated” arrows. Please review.

Response: Thank you so much. We would like to thank you for your thoughtful and sincere review. We have revised our Figure 3 according to you recommendation.

Line 328 Figure 4 - I would suggest replacing “attack” with “challenge” or “infection”. I would also suggest having the Y-axis scale the same for Panels D, E, & F and also for Panels G, H, and I – this would enable easier comparison of these results.

Response: Thank you very much. We would like to thank you for your thoughtful and sincere review. We have revised our Figure 4 according to you recommendation.

Round 2

Reviewer 2 Report

The authors have adequately addressed the questions and comments I made during my review of version 1 of their manuscript.

Minor suggestions:

Line 19 suggest replacing "make sure" with "evaluate"

Line 397 suggest revision "were not significantly different"

Line 398 Please revise "normal group" - for clarity, the authors should only use the group names shown in Table 1.
